

# A quantum annealing protocol to solve the nuclear shell model

Emanuele Costa[1,2], Axel Pérez-Obiol[3], Javier Menéndez[1,2],
Arnau Rios[1,2], Artur García-Sáez[4,5] and Bruno Juliá-Díaz[1,2]

**1** Departament de Física Quàntica i Astrofísica,
Universitat de Barcelona, 08028 Barcelona, Spain
**2** Institut de Ciències del Cosmos, Universitat de Barcelona, 08028 Barcelona, Spain
**3** Departament de Física, Universitat Autònoma de Barcelona, 08193 Bellaterra, Spain
**4** Barcelona Supercomputing Center, 08034 Barcelona, Spain
**5** Qilimanjaro Quantum Tech, 08019 Barcelona, Spain

## Abstract

The nuclear shell model accurately describes the structure and dynamics of atomic nuclei. However, the exponential scaling of the basis size with the number of degrees of freedom hampers a direct numerical solution for heavy nuclei. In this work, we present a quantum annealing protocol to obtain nuclear ground states. We propose a tailored driver Hamiltonian that preserves a large gap and validate our approach in a dozen nuclei with basis sizes up to $10^5$ using classical simulations of the annealing evolution. We explore the relation between the spectral gap and the total time of the annealing protocol, assessing its accuracy by comparing the fidelity and energy relative error to classical benchmarks. While the nuclear Hamiltonian is non-local and thus challenging to implement in current setups, the estimated computational cost of our annealing protocol on quantum circuits is polynomial in the single-particle basis size, paving the way to study heavier nuclei.

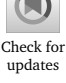
# 1 Introduction

Neutrons and protons, collectively known as nucleons, bind together through the strong interaction to form atomic nuclei: quantum many-body systems that exhibit a plethora of interesting phenomena. While the properties of nuclear ground states in stable nuclei are generally understood, experimental evidence has often challenged some fundamental tenets of nuclear structure. For instance, the cornerstones of nuclear structure are magic numbers, specific numbers of protons and neutrons associated with particularly bound, loosely correlated, spherical nuclei. However, the evolution of magic numbers in unstable nuclei challenges the current understanding of nuclear forces [1–4]. Similarly, the existence of short-range correlations between nucleon pairs with definite isospin is somewhat at odds with the traditional picture of the low-lying structure of nuclei [5]. In addition, the coexistence at low energies of nuclear states associated with different shapes is an elusive quantum property that requires a solid theoretical underpinning [6–8]. The properties of nuclear structure are also relevant beyond nuclear physics, as they impact astrophysics [9, 10] and pave the way for the search for new physics beyond the standard model of particle physics [11, 12].

In the context of nuclear theory, the nuclear shell model (NSM) [13–16] plays a key role in describing the structure of nuclei. In analogy to configuration-interaction (CI) methods in quantum chemistry, a NSM description separates nuclear configurations into distinct components. An inert space with magic numbers of neutrons and protons acts as a core, assumed to decouple from the rest of the system by a relatively large energy gap. All the dynamics is instead contained in neutrons and protons occupying a series of valence orbitals. The diagonalization of many-body states in the NSM valence space enables detailed studies of nuclear ground and excited states, as well as strong, weak, and electromagnetic transitions, among other properties. Unfortunately, the NSM, like any other CI method, is affected by the exponential scaling of the many-body Hilbert space with the number of valence nucleons. This limitation restricts the classical simulation of heavy nuclei, particularly those far from magic numbers. However, insights from quantum information have opened new paths for studying nuclei using classical algorithms [17–28].

Quantum computing, algorithms, and devices are promising tools for studying quantum many-body systems. Two distinct types of approaches have been used to simulate ground states. On one hand, variational quantum eigensolvers (VQEs) [29–33] should provide an advantage in the Noisy Intermediate-Scale Quantum era. On the other hand, quantum annealers [34–36] offer a more physics-inspired approach to quantum simulation. In either approach, simulations have been applied not only to condensed matter systems [37–40], but also to hadron and nuclear physics [41, 42] and quantum chemistry [43–47].

Several strategies have been proposed to solve the NSM and obtain the properties of nuclei using quantum algorithms [48–57]. A promising avenue, also applied in other nuclear structure methods [58–60], is that of the ADAPT-VQE algorithm [44], a VQE with an adaptive ansatz that tailors the algorithm to the corresponding energy landscape in a very efficient way. While classical simulations are encouraging, actual implementations may suffer from measurement noise or the presence of local minima in the classical optimization [61–63]. This suggests the need to explore alternative methods based on different ideas.

Quantum annealing (QA) is a method used to find the ground state of Hamiltonians through time evolution. Recently, QAs [64, 65] have been widely used to solve quadratic unconstrained binary optimization (QUBO) problems [66, 67]. With QA, one aims to find the solution of this classical NP-hard problem by encoding it in the ground state of an Ising Hamiltonian. This is achieved via time evolution; however, the small spectral gap of the time-dependent Hamiltonian hampers the method's effectiveness. Several techniques have been developed to improve the robustness of the annealing, such as tailored driver Hamiltonians

[68, 69] or adaptive time schedules [70]. Several studies have been also focused on both the criteria of convergence [71–73] and the speed-up of the QA protocol via counteradiabatic drivings [74, 75]. Moreover, the application range for QA has expanded beyond QUBO problems to include the ground state of condensed matter systems [39, 40] and molecules [43, 47, 76, 77], yielding promising results.

In this paper, we investigate the implementation of a QA protocol to find the ground state for NSM Hamiltonians. QA is based on the time evolution under a time-dependent Hamiltonian and, unlike VQEs, it does not require any classical optimization. In order to assess the feasibility of the method, we introduce a novel driver Hamiltonian, specifically designed for the NSM. We emphasize that the target NSM Hamiltonian is non-local, and therefore, implementing our QA protocol using current setups may be unfeasible. To understand the applicability of the method in existing quantum devices, we study the computational cost of implementing the protocol in a quantum circuit, utilizing a digitalized Trotter decomposition of the time evolution [39, 70].

The paper is organized as follows. In Sec. 2, we present the NSM and introduce the nuclei where we test the QA protocol. This is further elaborated in Sec. 3, where we also discuss the metrics used to benchmark the accuracy of the method. Section 4 presents the driver Hamiltonian employed in our simulations. Section 5 then turns to addressing the time evolution of our QA protocol and its accuracy for three different time intervals. In Sec. 6, we estimate the computational cost of a QA implementation on a quantum circuit using a digitalized Trotter decomposition of the time evolution. Finally, conclusions and outlook are presented in Sec. 7.

## 2 Nuclear shell model Hamiltonian

The NSM [13–16] describes nuclear structure as an assortment of nucleons occupying shells with different energies. In the NSM picture, the nucleons that contribute to the dynamics are just those in the valence shell. In this way, it is possible to confine the configuration space of the nucleon modes to this shell, tracing out the degrees of freedom related to the core and the higher energy shells. Each isotope with mass number $A = N + Z$ is thus split into a core with magic numbers, $N_c$ and $Z_c$, and the corresponding valence neutrons and protons, with numbers $N_n$ and $Z_p$ respectively.

Since the nuclear force is rotationally invariant, each nucleon mode or single-particle orbital is described by the quantum numbers

$$|a\rangle = |n, l, j, m, t_z, t\rangle \,, \tag{1}$$

where $n$ is the principal quantum number; $l$ is the orbital angular momentum; $j$, the total angular momentum of the nucleon; $m$, the projection of the total angular momentum; and $t$ and $t_z$ are the isospin and the isospin projection, respectively. In this representation, the target NSM Hamiltonian is written as

$$\hat{H}_T = \sum_a \varepsilon_a c_a^\dagger c_a + \frac{1}{4} \sum_{abcd} \bar{v}_{abcd} c_a^\dagger c_b^\dagger c_d c_c \,, \tag{2}$$

where $c_a^\dagger$ and $c_a$ are the fermionic creation and annihilation operators, $\varepsilon_a$ is the single particle energy of the nucleon mode $|a\rangle$ and $\bar{v}$ is the antisymmetrized two-body matrix. A nuclear shell has a set $J_s$ of allowed $j$ values (eg $j = 3/2$ and $j = 1/2$ in the $p$ shell, so $J_s = \{3/2, 1/2\}$). The number of nucleon modes per species is then $D = \sum_{j \in J_s}(2j + 1)$.

The two-body matrix elements $\bar{v}_{abcd}$ may be obtained from an effective field theory of nuclear forces, but often also include phenomenological adjustments [16, 78–81]. They are typically expressed in terms of the $M$ and $T_z$ independent — due to rotational and isospin symmetry — matrix elements in the coupled basis, $V_{abcd}(J, M, T, T_z) = \langle a, b, J, M, T, T_z | \hat{V} | c, d, J, M, T, T_z \rangle$,

Table 1: Nuclei studied with our QA protocol. $N_n$ and $Z_p$ are the valence neutron and proton numbers and $\dim \mathcal{F}_0$ is the dimension of the $M = 0$ subspace. In $^{28}$Si, nucleons occupy half of the possible orbitals—modes—in the valence space, and thus $\dim \mathcal{F}_0$ within the corresponding shell is maximum.

| Nucleus | $N_n$ | $Z_p$ | Shell | $\dim \mathcal{F}_0$ |
|---------|-------|-------|-------|----------------------|
| $^8$Be  | 2 | 2 | $p$ | 51 |
| $^{10}$Be | 4 | 2 | $p$ | 51 |
| $^{12}$Be | 6 | 2 | $p$ | 5 |
| $^{12}$C | 4 | 4 | $p$ | 51 |
| $^{18}$O | 2 | 0 | $sd$ | 14 |
| $^{20}$O | 4 | 0 | $sd$ | 81 |
| $^{22}$O | 6 | 0 | $sd$ | 142 |
| $^{20}$Ne | 2 | 2 | $sd$ | 640 |
| $^{22}$Ne | 4 | 2 | $sd$ | 4206 |
| $^{24}$Ne | 6 | 2 | $sd$ | 7562 |
| $^{24}$Mg | 4 | 4 | $sd$ | 28503 |
| $^{26}$Mg | 6 | 4 | $sd$ | 51630 |
| $^{28}$Si | 6 | 6 | $sd$ | 93710 |
| $^{30}$Si | 8 | 6 | $sd$ | 51630 |
| $^{32}$Ar | 6 | 10 | $sd$ | 7562 |

where $J$, $M$, $T$ and $T_z$ are two-body angular momentum and isospin quantum numbers. The two bases are related by Clebsch-Gordan coefficients, $\mathcal{C}$ [82],

$$
\begin{aligned}
\bar{v}_{abcd} = \sum_{J,M,T,T_z} & \left[ \frac{\sqrt{1-\delta_{ab}(-1)^{J+T}}}{1+\delta_{ab}} \frac{\sqrt{1-\delta_{cd}(-1)^{J+T}}}{1+\delta_{cd}} \right]^{-1} \\
& \times \mathcal{C}(J,M,j_a,m_a,j_b,m_b) \mathcal{C}(J,M,j_c,m_c,j_d,m_d) \\
& \times \mathcal{C}(T,T_z,t_a,t_{z,a},t_b,t_{z,b}) \mathcal{C}(T,T_z,t_c,t_{z,c},t_d,t_{z,d}) \\
& \times \mathcal{V}_{abcd}(J,M,T,T_z).
\end{aligned}
\tag{3}
$$

In our study, we consider the Cohen-Kurath interaction (CKI) in the $p$ shell [83], and the universal $sd$ interaction (USDB) in the $sd$ shell [84]. Since the standard nuclear units for energies are MeV—the $\bar{v}_{abcd}$'s are order $o(1)$ in these units—throughout this work we set the scale $\omega = 1$ MeV. All our results are given in terms of this energy scale.

The Fock space $\mathcal{F}$ of the NSM Hamiltonian is spanned by a many-body basis $\mathcal{B}$, composed of Slater determinants $|\mathbf{s}\rangle$ of the nucleon mode orbitals $|a\rangle$. These many-body states are eigenstates of the total angular momentum projection $M$ and the total isospin projection $T_z$. Moreover, a Slater determinant can be expressed as a bitstring

$$
|\mathbf{s}\rangle = |\{s[i]\}_{i \in [1,2D]}\rangle = |n_0, n_1, ..., n_{2D}\rangle \,,
\tag{4}
$$

with $s[a]$ being the $a$-th component of the classical bitstring $\mathbf{s}$ and $n_a \in \{0,1\}$. The dimension of $\mathcal{F}$ is given by

$$
\dim \mathcal{F} = \binom{D}{N_n} \times \binom{D}{Z_p} \,,
\tag{5}
$$

which increases as the number of occupied valence orbitals approaches mid-shell. When this occurs, numerical simulations in heavy nuclei become challenging. One way to mitigate this

is to exploit symmetries. For instance, due to rotational invariance, we can restrict the Fock space to lie in the $M = 0$ subspace, with the corresponding many-body basis $\mathcal{B}_0$. Table 1 reports the dimension of the $M = 0$ subspace, $\dim \mathcal{F}_0$, for all nuclei studied in this work. We specifically focus on nuclei with even $N_n$ and $Z_p$, which have a nondegenerate ground state with $J = 0$ and $M = 0$. While some of these systems have been simulated with VQEs in previous works [52–55], those with the largest basis sizes $\dim \mathcal{F}_0 > 2 \cdot 10^4$ are addressed in this work for the first time using a quantum algorithm.

## 3 Quantum annealing

QA [85, 86] is a method for obtaining the ground state of Hamiltonians through "slow" time evolutions. The time evolution is defined by an interpolating Hamiltonian

$$\hat{H}(t) = [1 - \lambda(t)]\hat{H}_D + \lambda(t)\hat{H}_T, \tag{6}$$

where $\hat{H}_T$ is the Hamiltonian that represents the target ground state and $\hat{H}_D$ is the so-called driver Hamiltonian. The time schedule $\lambda(t)$ is such that $\lambda(t = 0) = 0$ and $\lambda(t = \tau) = 1$ with $t \in [0, \tau]$. In this exploratory study, we consider a linear time schedule $\lambda(t) = t/\tau$, although more sophisticated schemes exist [70]. The driver Hamiltonian generates quantum fluctuations in the system because $[\hat{H}_T, \hat{H}_D] \neq 0$.

The time evolution is described by the Schrödinger equation

$$i\frac{d}{dt}|\Psi(t)\rangle = \hat{H}(t)|\Psi(t)\rangle, \tag{7}$$

where one takes the initial condition $|\Psi(0)\rangle = |\Psi_D\rangle$ as the ground state of the driver Hamiltonian $\hat{H}_D$. In general, $\hat{H}_D$ is chosen such that $|\Psi_D\rangle$ is a trivial state, like a product state representing a classical bitstring. If the time schedule is slow enough (long $\tau$ limit), the time evolution of the state $|\Psi(t)\rangle$ follows the instantaneous ground state $|e_0(t)\rangle$ of $\hat{H}(t)$. Other instantaneous eigenstates of the interpolating Hamiltonian,

$$\hat{H}(t)|e_i(t)\rangle = e_i(t)|e_i(t)\rangle, \tag{8}$$

may be active during the time evolution. In order to approximately satisfy the adiabatic condition [34, 87], the time evolution should last $\tau \gg \tau_\Delta$, with

$$\tau_\Delta \propto O\left(\Delta^{-2}\right), \tag{9}$$

where $\Delta = \min_t \left(e_r(t) - e_0(t)\right)$ is the minimum gap spectrum and $r$ indicates the first excited level that goes in resonance with the evolved state. Therefore, the feasibility of QA, in short time intervals, strongly depends on $\Delta$. However, for a large system, obtaining information about the spectral gap is challenging [88] and QA may be the only tool available to investigate $\Delta$.

To benchmark the feasibility of QA in NSM Hamiltonians, we study both the final fidelity,

$$F(\tau) = |\langle \Psi(\tau)|\Psi_T\rangle|^2, \tag{10}$$

where $|\Psi_T\rangle$ is the target ground state, and the final energy relative error with respect to the target ground state energy, $E_T$,

$$\Delta_r e(\tau) = \frac{e(\tau) - E_T}{|E_T|} = \frac{\langle \Psi(\tau)|\hat{H}_T|\Psi(\tau)\rangle - E_T}{|E_T|}, \tag{11}$$

where $e(\tau)$ is the energy at the end of the protocol. We compute both $|\Psi_T\rangle$ and $E_T$ using classical benchmarks. Both $F(\tau)$ and $\Delta_r e(\tau)$ quantify the quality of the QA algorithm in reaching the true target ground state, achieved exactly if $F(\tau) = 1$ and $\Delta_r e(\tau) = 0$. A useful property for quantifying the adiabatic nature of the evolution is the population of the instantaneous eigenstates,

$$p_i(t) = |\langle \Psi(t) | e_i(t) \rangle|^2, \tag{12}$$

which ideally is $p_0(\tau) = 1$ and $p_i(\tau) = 0$ for $i > 0$. Non-zero intermediate-time populations, $p_i(t) \neq 0$, indicate which levels contribute the most to the nonadiabatic dynamics.

## 4 Driver Hamiltonian

As mentioned in Sec. 3, the driver Hamiltonian generates quantum fluctuations in the system because it does not commute with the target Hamiltonian. Moreover, since it corresponds to the initial Hamiltonian $\hat{H}(0)$, it should have a ground state simple to encode, such as a classical state. In the context of QA [89], $\hat{H}_T$ is typically a classical spin Hamiltonian, while the common driver Hamiltonian $\hat{H}_D$ is a transverse field, ideal for introducing quantum fluctuations to the system.

However, in the NSM framework, the situation is different. On the one hand, we want to conserve both the total number of particles and the symmetries of the target Hamiltonian, in order to restrict the dynamics to a physical subspace of the Fock space. On the other hand, we need a simple Hamiltonian whose ground state can be codified by a classical state, such as an element of the many-body basis of $\mathcal{F}$. With this in mind, we consider as the initial many-body state the one that occupies the lowest-energy single-particle orbitals. Next, the degenerate orbitals with different $m$ are monotonically occupied, starting from the highest $|m|$. For example, in $^{20}$O, with $N_n = 4$ and $Z_p = 0$, we have

$$|\tilde{\mathbf{s}}\rangle = c^\dagger_{a=(5/2,5/2)} c^\dagger_{b=(5/2,3/2)} c^\dagger_{c=(5/2,-3/2)} c^\dagger_{d=(5/2,-5/2)} |\text{Vac}\rangle, \tag{13}$$

where $|\text{Vac}\rangle$ represents the nuclear core vacuum state, and we indicate only the quantum numbers $(j, m)$ of each valence neutron mode. That is, our protocol complements the natural order of filling the nucleon modes with a specific order according to $m$.

The expectation value of the Hamiltonian for this initial state is

$$E_0 = \langle \tilde{\mathbf{s}} | \hat{H}_T | \tilde{\mathbf{s}} \rangle. \tag{14}$$

We choose the driver Hamiltonian corresponding to $|\tilde{\mathbf{s}}\rangle$ as

$$\hat{H}_D = \frac{E_0}{(N_n + Z_p)} \sum_a \tilde{s}[a] \hat{n}_a, \tag{15}$$

where $\hat{n}_a = c^\dagger_a c_a$ and $\tilde{\mathbf{s}} = (\tilde{s}[a_0], \tilde{s}[a_1], \tilde{s}[a_2], ..., \tilde{s}[a_{2D}])$ is the bitstring related to $|\tilde{\mathbf{s}}\rangle$ as described in Eq. (4). The left panel of Fig. 1 shows the matrix elements of the driver Hamiltonian in the single-particle basis (nucleon modes) for $^8$Be. At $t = 0$, the Hamiltonian is diagonal in both the single-particle and many-body bases. The labeling of the single-particle basis in the $p$ shell follows the convention provided in the bottom panel. Our choice of $\hat{H}_D$ fulfills

$$[\hat{H}_T, \hat{H}_D] \neq 0, \tag{16}$$

hence guaranteeing the build-up of quantum fluctuations. We define the set of many-body states with M=0 projection $\mathcal{B}_0 = \{|\mathbf{s}\rangle \in \mathcal{B} \mid \hat{M} |\mathbf{s}\rangle = 0\}$, indicating the many-body basis with $\mathcal{B}$. As a consequence, due to the fact that

$$[\hat{H}_D, \hat{M}] = 0, \tag{17}$$

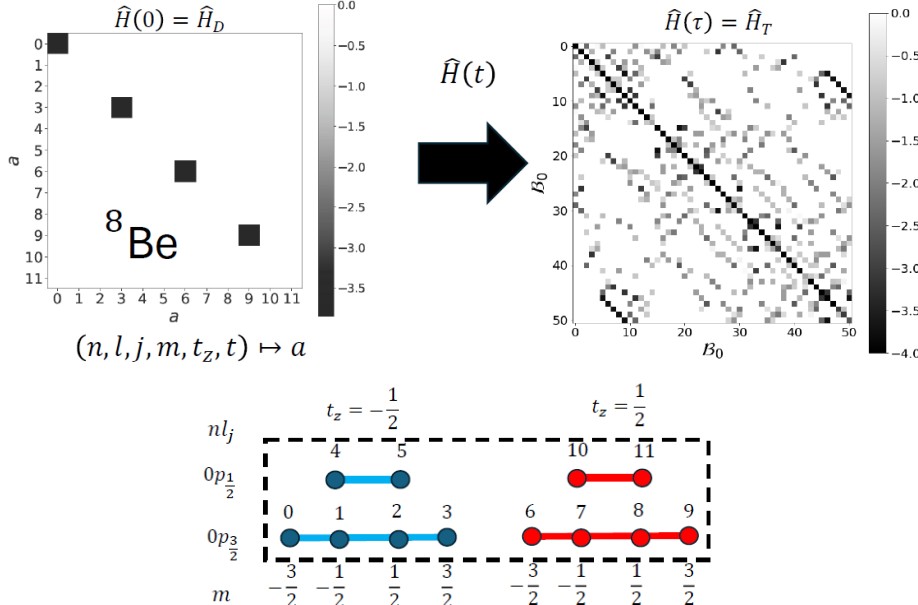

Figure 1: Left panel: matrix elements of the driver Hamiltonian $\hat{H}_D$ in the single-particle basis (nucleon modes) for $^8$Be in the $p$ shell. The bottom panel indicates the ordering of the modes, where left (right) single-particle levels correspond to neutrons (protons). Right panel: after the annealing protocol is finished, the Hamiltonian $\hat{H}(t)$ becomes the target Hamiltonian $\hat{H}(\tau) = \hat{H}_T$, which is significantly non-local in the many-body basis.

the evolution is confined to the $M = 0$ sector. Finally, it is noteworthy that our choice for the initial state corresponds to the minimum energy Slater determinant in $\mathcal{B}_0$

$$|\mathbf{s}_{min}\rangle = \arg \min_{|\mathbf{s}\rangle \in \mathcal{B}_0} \langle \mathbf{s}| \hat{H}_T |\mathbf{s}\rangle \,, \tag{18}$$

in all studied nuclei except $^{20}$O, $^{22}$Ne, $^{24}$Mg and $^{26}$Mg. A lower energy closer to the ground-state energy of the target Hamiltonian may be advantageous for a potential QA realization, although we emphasize that it does not guarantee faster or more efficient convergence than other initial states [90, 91].

During the time evolution induced by the annealing protocol, the Hamiltonian $\hat{H}(t)$ becomes progressively non-local in the many-body basis. The right panel of Fig. 1 shows the target Hamiltonian for $^8$Be, which exhibits a substantial non-diagonal structure. Indeed, the off-diagonal entries of $\hat{H}(t)$ are largest at $t = \tau$, when $\hat{H}(\tau) = \hat{H}_T$.

# 5 Simulations of nuclei with a quantum annealing protocol

We now turn to analyzing the results of our QA protocol. To simulate the time evolution, we evolve from an initial time $t_0 = 0$ to a final time $\tau = N_t \Delta t$, assuming a uniform time spacing such that $\Delta t \omega = 0.1$. We decompose the unitary time evolution operator using sparse matrix exponentiation from the Scipy library [92]

$$\hat{U}(\tau, t_0) = \prod_{k=0}^{N_t} \hat{U}_k = \prod_{k=0}^{N_t} \exp\left[-ik\Delta t \hat{H}(t_k)\right], \tag{19}$$

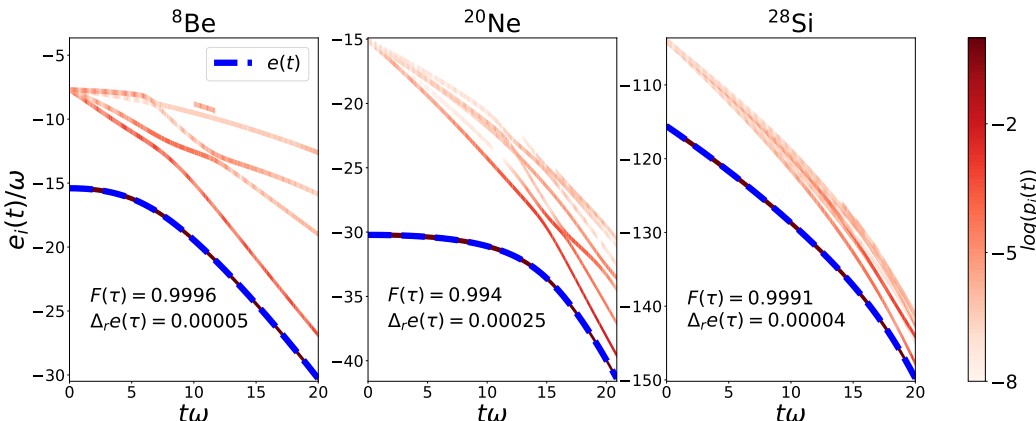

Figure 2: Adiabatic evolution for $^8$Be (left), $^{20}$Ne (central), and $^{28}$Si (right panel). The lines represent the instantaneous energies $e_i(t)$ associated to $\hat{H}(t)$ with $i = 0, \cdots, 9$. The dashed blue line shows the expectation value of the energy $e(t) = \langle \Psi(t) | \hat{H}(t) | \Psi(t) \rangle$, and the colorbar indicates the population magnitude of each state in terms of $\log(p_i(t))$. The oscillations in population signal non-adiabatic effects in the dynamics. We show in each panel the value of the final fidelity, $F(\tau)$, as well as the final relative energy error with respect to the exact benchmark, $\Delta_r e(\tau)$. We achieve $\Delta_r e(\tau) < 10^{-4}$ for the three nuclei, with $^{28}$Si showing the best performance in spite of being a mid-shell nucleus. The energy scale is $\omega = 1$ MeV.

where $N_t \in [100, 300]$ is the number of time steps. We compute the instantaneous energy levels of $\hat{H}(t)$, $e_i(t)$, and the corresponding eigenstates $|e_i(t)\rangle$ using exact diagonalization.

We start our analysis by simulating the adiabatic evolution of $^8$Be, $^{20}$Ne, and $^{28}$Si. The three panels of Figure 2 present the time evolution of the ground-state energies for these isotopes. We focus on a final time $\tau\omega = 20$. Each panel in Fig. 2 shows, with a dashed blue line, the evolution of the expectation value $e(t) = \langle \Psi(t) | \hat{H}(t) | \Psi(t) \rangle$. In addition, we show the 10 instantaneous lowest-energy eigenstates as a function of time, $e_i(t)$. We highlight the population magnitude $p_i(t)$ of each one of these states with a colour bar, such that states close to the full population have intense colours and unoccupied states appear transparent. Each panel also quantifies the quality of the QA protocol in terms of the final fidelity $F(\tau)$ in Eq. (10) and the relative energy error $\Delta_r e(\tau)$ in Eq. (11).

For the three nuclei, we find that the expectation value $e(t)$ exactly follows the instantaneous ground-state energy, $e_0(t)$. This agreement indicates the quality of the annealing protocol, which may be considered adiabatic by this measure. The energy difference between the initial $e(t = 0)$ and the final energy $e(\tau)$ is of the order of a few tens of MeV. For $^8$Be and $^{20}$Ne, the energy remains relatively constant initially and, after some time, decreases linearly and more steeply until reaching the final value. In contrast, for $^{28}$Si we observe two linearly decreasing regimes with different slopes.

In our protocol, the time evolution of the 9 excited states differs significantly from that of the ground state. At $t = 0$, the first 9 excited states are degenerate. At this initial stage, there is a considerable gap, in the range between $5 - 20$ MeV, between the ground state and the excited states. For $t \in ]0, \tau[$, the excited levels break this degeneracy and evolve over time in a complex pattern. Several level crossings occur, particularly among the more energetic states. As $t$ approaches the final value, however, one distinct level consistently edges closer to the ground state. This level also provides the minimum gap, $\Delta$, very close to—if not at—the end of the time evolution, $t \simeq \tau$. The minimum gap is of the order of $2 - 5$ MeV and likely corresponds to the spectral gap between the ground and first excited state in the $M = 0$ subspace of each nucleus.

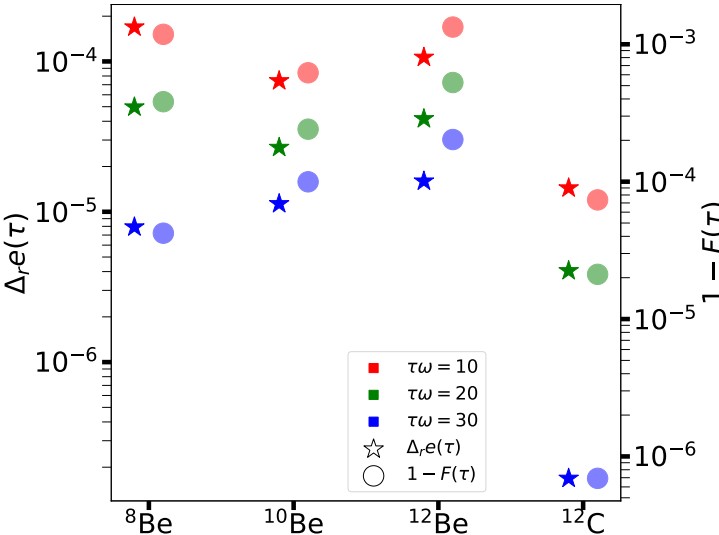

Figure 3: Relative energy error, $\Delta_r e(\tau)$ (stars, left axis), and infidelity, $1-F(\tau)$ (circles, right axis), for $p$-shell nuclei at the end of our protocol, for timescales $\tau\omega = 10$ (red symbols), $\tau\omega = 20$ (green) and $\tau\omega = 30$ (blue).

Our findings suggest that the evolution of the excited states may vary significantly from nucleus to nucleus. The left and central panels, corresponding to $^8$Be and $^{20}$Ne, show notable crossings at timescales comparable to the steeper changes in slope of $e(t)$. In contrast, for $^{28}$Si, the only case where the driver Hamiltonian fully occupies the $j = 5/2$ subshell, the excited states appear much closer together over time and exhibit less significant structure.

In addition to the instantaneous energies, Fig. 2 illustrates the population $p_i(t)$ of the instantaneous excited levels. Oscillations in these populations indicate complex dynamical processes that are not adiabatic, allowing us to gauge non-adiabatic effects based on the size of these populations. For both $^8$Be and $^{28}$Si, these never exceed $p_i(t) < 10^{-3}$. In contrast, non-adiabatic effects are larger for $^{20}$Ne, which reaches $p_1(t) \sim 5 \cdot 10^{-3}$. In an ideal QA protocol, the time-evolved state would be fully transferred into the ground state of the target Hamiltonian. The presence of excited state population in the final state thus indicates that the transfer is not fully adiabatic. In agreement with these qualitative expectations, the final infidelity is relatively close for $^8$Be and $^{28}$Si, $1-F(\tau) \sim 10^{-4}$, whereas for $^{20}$Ne it is about an order of magnitude larger, $1-F(\tau) \sim 10^{-3}$.

To go further in the analysis, we explore the QA protocol for all nuclei in Table 1. Figure 3 presents, for $p$-shell nuclei, the relative error of the final energy ($\Delta_r e$, stars, left axis) and the infidelity ($1-F(\tau)$, circles, right axis) for three different final times $\tau\omega \in \{10, 20, 30\}$ using a logarithmic scale. This allows us to assess the significance of the timescale in the QA protocol. Figure 4 shows similar results for $sd$-shell nuclei.

There are several interesting features in Fig. 3. First, the final results follow the expected trend in terms of $\tau$ dependence: the slower the QA protocol is—meaning longer $\tau$—the better the results are regarding infidelity and relative energy. The errors in the latter are in the range $\Delta_r e(\tau) \in [8 \cdot 10^{-6}, 2 \cdot 10^{-4}]$, whereas for infidelities $1-F(\tau) \in [6 \cdot 10^{-7}, 10^{-3}]$. The best accuracies correspond to $\tau\omega = 30$ and the worst to $\tau\omega = 10$. Whereas for $^8$Be the relative energy error (infidelity) is about $10^{-4}$ ($10^{-3}$) for $\tau\omega = 10$, it reduces by more than an order of magnitude to $9 \cdot 10^{-6}$ ($4 \cdot 10^{-5}$) for $\tau\omega = 30$. Likewise, a clear improvement of about two orders of magnitude is found for $^{12}$C. In contrast, the results for $^{10}$Be or $^{12}$Be (an isotope with a very small Fock space dimension of dim $\mathcal{F}_0 = 5$) show a milder improvement with $\tau$ of less than an order of magnitude for each quantity.

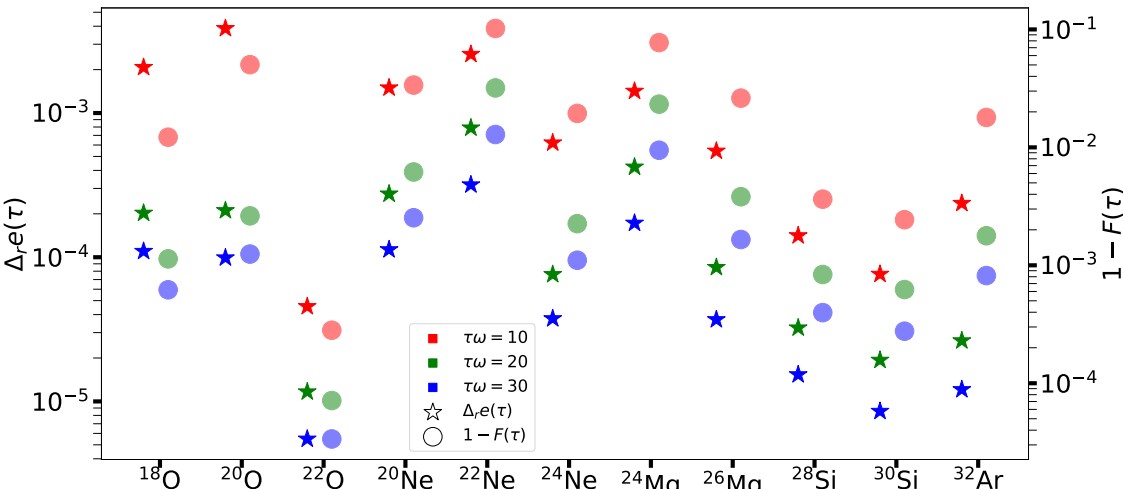

Figure 4: Same as Fig. 3 for $sd$-shell nuclei. For $\tau\omega = 30$, all nuclei reach an infidelity $1-F(\tau) < 2\cdot 10^{-2}$.

For the $sd$ shell, the picture becomes even more interesting. Figure 4 shows, in analogy to the $p$-shell results of Fig. 3, the final energy relative errors and infidelities for the $sd$-shell nuclei in Table 1, considering three different adiabatic timescales. We observe a larger variation compared to the $p$-shell results in both the left (energy) and right (infidelity) vertical axes. Just as in the case of the $p$ shell, Fig. 4 shows that in all cases extending the duration of the adiabatic protocol improves the QA performance.

In this shell, we identify three subsets of nuclei with different behaviors. The first subset includes $^{20}$Ne, $^{22}$Ne, and $^{24}$Mg. These nuclei have relative energies (infidelities) of the order $10^{-3}$ $(10^{-1})$ for $\tau\omega = 10$, which improve mildly until reaching relative energies in the vicinity of $10^{-4}$. The infidelities for $\tau\omega = 30$, in contrast, span a broader range, between $10^{-2}$ and $10^{-3}$. For these nuclei, the proton and neutron modes in $\hat{H}_D$ do not fully occupy the lowest-energy $j = 5/2$ sub-shell. Thus, following our driver Hamiltonian scheme described in Sec. 4, the modes with lowest $m = \pm 1/2$ remain unoccupied. A rearrangement of this initial choice may improve the accuracy of the QA protocol.

The QA results are better for a second subset of nuclei, $^{18}$O, $^{20}$O, $^{32}$Ar, $^{24}$Ne and $^{26}$Mg, with energy errors at $\tau\omega = 30$ in the range $\Delta_r e(\tau) \in [10^{-5}, 10^{-4}]$ and infidelities $1 - F(\tau) \in [6\cdot 10^{-4}, 2\cdot 10^{-3}]$. Moreover, for these nuclei, we observe faster convergence with the time scale $\tau$, especially between $\tau\omega = 10$ and $\tau\omega = 20$. Here, the nucleon modes occupied in the driver Hamiltonian correspond to a closed subshell in either the neutron or proton valence space. In particular, $^{18}$O and $^{20}$O have a closed proton $p$ shell, whereas $^{32}$Ar, $^{24}$Ne and $^{26}$Mg have the $j = 5/2$ subshell completely filled.

The third subset of nuclei includes $^{22}$O, $^{28}$Si and $^{30}$Si. These nuclei show the best QA accuracy. This is somewhat unexpected, as $^{28}$Si has the largest many-body basis among all considered nuclei, $\dim \mathcal{F}_0 \simeq 10^5$, see Table 1. At $\tau\omega = 30$, $^{22}$O reaches $\Delta_r e(\tau) \sim 5\cdot 10^{-6}$ and $1 - F(\tau) \sim 5\cdot 10^{-5}$, $^{28}$Si simulations yield $\Delta_r e(\tau) \sim 2\cdot 10^{-5}$ and $1 - F(\tau) \sim 3\cdot 10^{-4}$, while $^{30}$Si gets $\Delta_r e(\tau) \sim 8\cdot 10^{-6}$ and $1 - F(\tau) \sim 3\cdot 10^{-4}$. These metrics surpass those obtained for any other nucleus. A common feature of these three nuclei is that the driver Hamiltonian completely fills subshells for both neutrons and protons. For neutrons, the $j = 5/2$ subshell is filled for $^{22}$O and $^{28}$Si and both the $j = 5/2$ and $j = 1/2$ subshells are closed for $^{30}$Si. For protons, oxygen has a closed $p$ shell, and for silicon, the $j = 5/2$ subshell is filled. We note that the closure of the $j = 5/2$ subshell corresponds to the magic number $N = 14$, leading to systems with suppressed nuclear correlations, which should be easier to simulate. In fact,

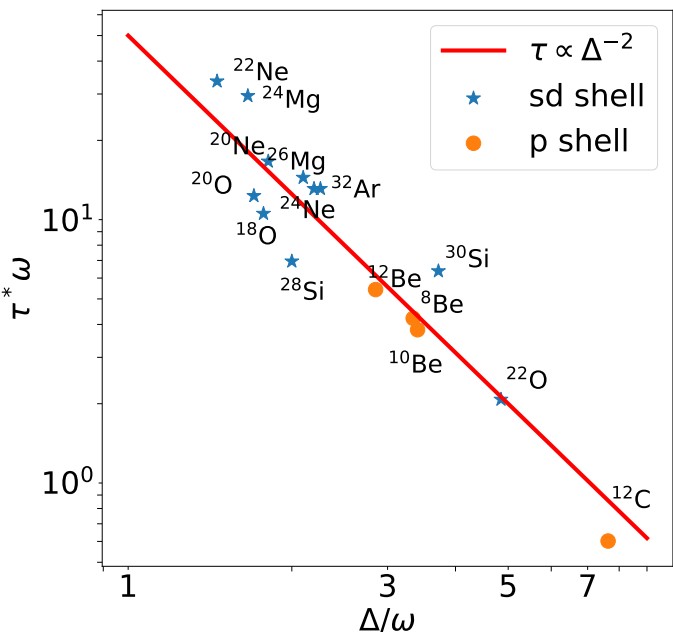

Figure 5: Time $\tau^*$ needed to reach a fidelity $F = 0.99$ with our QA protocol as a function of the minimum gap, $\Delta$, for all nuclei studied in this work, including $p$-shell (orange circles) and $sd$ shell isotopes (blue stars). Our results follow well the theoretical prediction $\tau \propto \Delta^{-2}$ (red curve). The energy scale is $\omega = 1$ MeV.

$^{22}$O, the nucleus with the best results, is doubly magic [93–95]. However, for $^{28}$Si, the doubly-magic configuration is not dominant [96, 97], as the ground state is a deformed state with sizable quadrupole correlations, also present in $^{30}$Si.

As mentioned in Sec. 4, the stability of the annealing protocol depends on the minimum gap $\Delta$ between the ground state and the lowest-energy state in resonance, $e_r(t)$. According to Eq. (8), systems with larger gaps are expected to reach better results with slower timescales. To identify the lowest-energy state in resonance, we examine the lowest $r > 0$ with $p_r(t) \neq 0$ through the evolution. We find that for all nuclei except $^{22}$O, $^{30}$Si and $^{12}$C—the three isotopes where our QA protocol achieves the best results—$e_r(t)$ corresponds to the first excited state. For $^{22}$O and $^{30}$Si, $e_r(t)$ is the second excited state, while for $^{12}$C, it is the third one. This occurs because some resonances are forbidden due to angular momentum. For example, in $^{12}$C, the first excited level $|e_1(t)\rangle$ remains in the $J = 1$ subspace throughout the evolution, whereas the ground state $|e_0(t)\rangle$ stays with $J = 0$.

In an effort to quantify these statements, Fig. 5 compares the final time $\tau^*\omega$ required to achieve a fidelity $F = 0.99$ with the corresponding gap $\Delta$. We consider a sequence of 100 adiabatic timescales $\tau_p$, with $p = 0, 1, 2, ..., 99$, in the range $\tau\omega \in [0.3, 40]$, running the QA protocol for these 100 timescales and selecting $N_t$ such that the time step $\Delta t\omega = 0.1$. We then identify the shortest timescale for a reasonable QA protocol, $\tau^*$, given by $p^*$ so that $\tau^* = \tau_{p^*}$ provides a fidelity $F^* > 0.99$.

Figure 5 presents the correlation between $\Delta$ and $\tau^*$ for all nuclei considered in this work, alongside the dashed red curve that indicates the approximate adiabatic condition $\tau \propto \Delta^{-2}$ [34]. $p$-shell isotopes are shown in solid circles, whereas $sd$-shell nuclei are displayed in blue stars. Our results fit the theoretical curve well, with a correlation coefficient $R^2 \simeq 0.89$. The discrepancies between the theoretical prediction and our results may arise because the gap alone does not fully capture non-adiabatic effects [98], and only offers an estimation of the robustness of the system.

Nonetheless, the range of minimum gaps observed is relatively narrow. Most $sd$-shell isotopes exhibit small gaps and long timescales, $\Delta \approx 2$ MeV and $\tau^*\omega = 7-30$, whereas $p$-shell nuclei cluster around $\Delta \approx 3$ MeV and $\tau^*\omega \approx 4-5$. Two nuclei show significantly larger gaps: $^{22}$O and $^{12}$C. Indeed, $^{12}$C shows the largest gap, a fact directly linked to its excited-state spectrum—$e_r$ is the third excited state. The corresponding $\tau^*$ value is the smallest of all nuclei, around $\tau^*\omega \approx 0.7$. These large gaps also explain the results in Figs. 3 and 4 which indicate that, for a fixed $\tau$, $^{12}$C and $^{22}$O are the best performing nuclei.

In general, our analysis suggests two robust features for QA in the NSM. Firstly, an increase in adiabatic timescales improves the results for all nuclei. A factor of 3 in timescale typically yields about one order of magnitude improvement in $\Delta_r e(\tau)$ and $1 - F(\tau)$. Secondly, the underlying shell structure of the different isotopes can have a non-trivial interplay with the adiabatic protocol, leading to larger minimum gaps than naively expected for some nuclei with closed subshells in the driver Hamiltonian. Overall, the dependence on neutron or proton number is not smooth and, in fact, some of the nuclei with the largest Fock space dimension can achieve exceptionally good accuracy with relatively fast evolutions.

Finally, we expect these main features of the QA protocol to be robust with respect to an enlargement of the Hilbert space for the same nucleus, which should mostly affect higher-energy excited states leaving the energy gap relatively stable. Likewise, we also anticipate similar results from using different nuclear Hamiltonians, as long as they capture well the experimental low-energy structure of the nuclei studied in this work. Indeed, we already explore two different sets of nuclear interactions for $p$- and $sd$-shell nuclei. These are phenomenological in nature, but the systematics of our results across different mass numbers suggest that our results do not depend significantly on the nuclear interaction used.

# 6 Computational cost of the quantum annealing protocol on a digital quantum computer

Unfortunately, because of the non-local nature of the NSM Hamiltonian, our QA protocol cannot be implemented on current QAs. The most direct implementation may be through a quantum circuit, using a Trotter decomposition [70, 99] of the unitary-time evolution operator, as presented in Eq. (19). Here we quantify the quantum resources associated with our QA protocol in a digital quantum computer.

An essential step is mapping fermions into qubits. As a starting point, we analyze the computational cost using the Jordan-Wigner mapping [100, 101]. In the qubit representation, the Hamiltonian can be written as

$$\hat{H}(t) = \sum_{i=1}^{R} \hat{h}_i(t) = \sum_{i=1}^{R} K_i(t) \hat{\sigma}_{a_{0,i}}^{\alpha_{a_{0,i}}} \hat{\sigma}_{a_{1,i}}^{\alpha_{a_{1,i}}} ... \hat{\sigma}_{a_{D,i}}^{\alpha_{a_{D,i}}}, \tag{20}$$

where $\hat{\sigma}^\alpha = \{\mathbb{I}, X, Y, Z\}$ are the Pauli operators, and $K_i(t)$ are the Hamiltonian matrix elements in the qubit basis. There is a total of $R$ matrix elements in this basis.

The full time evolution operator is the product of a series of $N_t$ unitary operators, $\hat{U}_k(t_k) = \exp[-ik\Delta t\hat{H}(t_k)]$, see Eq. (19). We can use a Trotter decomposition for this operator

$$\exp\left[-ir\Delta t\hat{H}(t_k)\right] \simeq \prod_{l=1}^{R} \exp\left[-ik\Delta t\hat{h}_l(t_k)\right], \tag{21}$$

in terms of the local operators, $\hat{h}_l(t_k)$, using the staircase algorithm to exponentiate the matrices [102]. To estimate the number of CNOT gates as a function of the number of nucleon modes per species $D$, we note that the number of two-body terms in the Hamiltonian is $\propto D^4$.

In the Jordan-Wigner mapping, each two-body term has an average Pauli string length of $\propto D/2$, and the number of CNOT gates required to exponentiate a Pauli operator of length $D/2$ is $(D-2)$ [102]. Therefore, the number of CNOTs for a unitary gate is

$$N_{\text{CNOT}} \propto D^4(D-2),\tag{22}$$

which means $N_{\text{CNOT}} \propto 10^4$ for the $p$ shell and $N_{\text{CNOT}} \propto 3 \cdot 10^5$ for the $sd$ shell, for a single timestep. By performing the Trotter decomposition using the OpenFermion library [103], we can directly count the number of CNOT gates needed for the implementation of the unitary time evolution gate in Eq. (21). For the $p$ shell, we obtain $N_{\text{CNOT}} = 4.8 \cdot 10^3$ and for the $sd$ shell, $N_{\text{CNOT}} = 1.1 \cdot 10^5$, confirming the estimates of Eq. (22) and their scaling. Nonetheless, the number of CNOT gates in the QA protocol could be further reduced by using alternative mappings that shorten the average Pauli operator length [39, 43, 77, 104–106], making the simulation more efficient. Circuit depths could also be reduced by improving the time schedule $\lambda(t)$ beyond the linear function used, or by implementing shortcuts to adiabaticity [107].

These numbers of CNOT gates—for a single timestep—are comparable to the total required for the convergence of the largest-dimension nucleus in the same valence space with ADAPT-VQE, $2 \cdot 10^3$ and $\sim 10^6$, respectively [52]. However, the QA CNOTs grow polynomially with $D$. In addition to that, we expect that the number of time steps also scales polynomially with $D$, since $\tau \propto \Delta^{-2}$, and $\Delta$ approaches the energy gap between the ground state and the first resonance state in $H_T$, that for nuclei scales polynomially with $D$ [13, 14]. In this perspective, our approach may scale better than ADAPT-VQE, which requires an exponentially growing number of CNOTs proportional to the number of many-body states [44, 52]. Hence, the polynomial scaling of the QA cost could offer an advantage for NSM calculations in heavier nuclei compared to the ADAPT-VQE method.

# 7 Conclusion

We present a QA protocol to obtain the ground states of atomic nuclei, suitable for quantum devices with high connectivity. Our work is, to our knowledge, the first application of quantum adiabatic computing to address a nuclear structure problem, enabling a clear synergy between two so-far disconnected research areas. In particular, we discuss a way to implement a driver Hamiltonian different from the ones used in the standard QA approach on spin systems. Our protocol is stable for all studied nuclei, both in the $p$ and $sd$ shells, encompassing systems with many-body basis dimensions up to $10^5$ states. Our QA approach achieves relative energy errors smaller than $O(10^{-4})$ in most cases, with fidelities greater than 0.99. The best performance is consistently found when using longer timescales, which correspond to slower annealing evolutions. Nonetheless, we notice that the scaling of the accuracy with the timescale varies between nuclei, with better scaling for those where the driver Hamiltonian fully occupies nucleon subshells. Additionally, we observe that the final time required to reach a high-quality fidelity, $\tau^*$, depends on the level gap, $\Delta$, in agreement with the adiabatic condition, $\tau \propto \Delta^{-2}$.

The interplay between nuclear structure and our QA protocol enhances results when the driver Hamiltonian completely fills nucleon subshells. This scenario can lead to large gaps—and thus improved performance—protected by angular momentum conservation, like in $^{12}$C or $^{22}$O. This highlights the non-smooth dependence of the QA accuracy with the number of nucleons observed throughout our study. In fact, some nuclei with larger many-body bases, such as $^{28}$Si, achieve the best QA results.

Due to the nonlocality of the NSM Hamiltonian, a direct implementation of the QA protocol using the Jordan-Wigner mapping on current QAs with low connectivity is not feasible. To compare this approach with state-of-the-art VQEs, we estimate the computational cost of the

QA protocol on a digital quantum device. While our method is more computationally expensive than the ADAPT-VQE algorithm for nuclei in the $p$ and $sd$ shells, it scales polynomially with respect to the number of nucleon modes because the scaling of the energy gap of the target Hamiltonian is also polynomial. For instance, for the nucleus with largest basis dimension we have studied, $^{28}$Si, the number of CNOT gates needed in our QA implementation is about one order of magnitude larger than with ADAPT-VQE. This suggests that QA could already offer advantages for heavier nuclei in the $pf$ shell.

This work establishes a foundation for using QA to study atomic nuclei. The quantum adiabatic framework for nuclear structure presented here opens several new potential research avenues at the interface between these two research areas. Future investigations could explore criteria of convergence of the QA protocol without recurring to classical simulations, based on the form of the scheduler or the structure of the driver Hamiltonian [71–73]. Moreover, it is possible to investigate ways to speed up the QA protocol by using counteradiabatic drivings [74, 75], studying the structure of the adiabatic gauge potential [98, 108] or using quantum optimal control techniques on the scheduler [70]. Future studies can also be focused on the qubit mappings that reduce the complexity of Pauli strings or assess QA implementations while considering the limitations of real quantum annealers. Additionally, applying the method to heavier nuclei in the $pf$ shell warrants further study to optimize the annealing protocol.

## Acknowledgments

E.C. acknowledges Rosario Fazio and Sebastiano Pilati for interesting discussions and suggestions.

**Funding information** This work has been financially supported by the Ministry of Economic Affairs and Digital Transformation of the Spanish Government through the QUANTUM ENIA project call - Quantum Spain project, and by the European Union through the Recovery, Transformation and Resilience Plan - NextGenerationEU within the framework of the Digital Spain 2026 Agenda. This work is also financially supported by MCIN/AEI/10.13039/501100011033 from the following grants: PID2020-118758GB-I00, PID2023-147475NB-I0 and PID2023-147112NB-C22; RYC2018-026072 through the "Ramón y Cajal" program funded by FSE "El FSE invierte en tu futuro"; CNS2022-135529 and CNS2022-135716 funded by the "European Union NextGenerationEU/PRTR", and CEX2019-000918-M to the "Unit of Excellence María de Maeztu 2020-2023" award to the Institute of Cosmos Sciences; and by the Generalitat de Catalunya, through grants 2021SGR01095 and 2021SGR00907.

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
