# Peer review of "A Quantum Annealing Protocol to Solve the Nuclear Shell Model"

_SciPost Physics, doi:SciPost Phys. 19, 062 (2025)_

## Round 2 · Referee Report · Anonymous (Referee 1) · 2025-1-28

Strengths
2-It gives a precise quantitative study
Weaknesses
2-Numerous Ccurrent studies of annealing are on the acceleration of adiabatic methods and/or on the proof of convergence. None of these two aspects are discussed properly in the article
Report
I have mixed feelings about the work. I feel this work is rather academic since we observe in practice what is expected, e.g.,. for instance, that the convergence property depends on the timescales tau omega. And, when two levels get closer to each other, the adiabaticity breaks down, as illustrated by the oscillation of the excited state occupation probabilities. Many groups today are trying to improve adiabatic methods to get faster convergence with the aim of being able to apply quantum platforms. If I am correct, this point is briefly mentioned only when quoting Ref. [100], while this is an active field today.
Also, the tests made in the article (relative energy errors, infidelity, …) rely on the possibility of creating the exact calculation back-to-back to the adiabatic technique. Assessing the convergence toward the proper target state is also an active field today in the context of quantum computing.
So, in short, the work seems a bit disconnected from current interests and discussions of practitioners trying to apply adiabatic methods on analog machines.
On the other hand, very few works have been made in nuclear physics using quantum annealing. As far as I know, this is one of the first attempts made to assess the quality of such a strategy for the nuclear shell model. From that point of view, the present work contains many interesting information and quantitative studies. It might even be seen as a milestone before trying to attack applications on real quantum platforms.
With this last argument, I estimate that the present article should be published provided that some more discussion is made on the recent efforts made to perform calculations using real quantum platforms, at least to better present the work as an academic intermediate step and quote the ongoing works that are made to fill the gap between this type of ideal work the progress that are made to apply quantum annealing today on real quantum processors.
Requested changes
1-Better discuss the current work on quantum annealing regarding the criteria of convergence and the different methods to accelerate the adiabatic approach.
Recommendation
Ask for minor revision

Author: Emanuele Costa on 2025-05-07 [id 5457]
(in reply to Report 1 on 2025-01-28)Dear Referee,
for the sake of clarity, we also report the pdf reply here in the box.
We thank Referee 1 for their careful reading of our manuscript, for the very positive assessment of
our work and for the very constructive comments and suggestions. Below, we address all of Referee
1’s recommendations in detail.
Referee 1, Comment and Requested Changes 1:
“With this last argument, I estimate that the present article should be published provided that some
more discussion is made on the recent efforts made to perform calculations using real quantum
platforms, at least to better present the work as an academic intermediate step and quote the ongoing
works that are made to fill the gap between this type of ideal work the progress that are made to
apply quantum annealing today on real quantum processors. [...] Better discuss the current work
on quantum annealing regarding the criteria of convergence and the different methods to accelerate
the adiabatic approach.”
Authors Response:
We thank Referee 1 for the suggestion. We certainly agree it is important to connect our work with
current efforts to perform simulations in real quantum annealers, including systems where classical
simulations are not feasible. To this end, we have added a new sentence in the Introduction:
• "Several studies have been also focused on both the criteria of convergence [71–73] and the
speed-up of the QA protocol via counteradiabatic drivings [74, 75]."
and an almost entire new paragraph in the Conclusion, indicating the various potential research
directions:
• "Future investigations could explore criteria of convergence of the QA protocol without recurring
to classical simulations, based on the form of the scheduler or the structure of the driver
Hamiltonian [71–73]. Moreover, it is possible to explore ways to speed up the QA protocol by
using counteradiabatic drivings [74, 75], studying the structure of the adiabatic gauge potential
[98, 108] or using quantum optimal control techniques on the scheduler [70]."
which include the following additional references incorporated into the manuscript:
• [71] Y. Kimura and H. Nishimori, Convergence condition of simulated quantum annealing for
closed and open systems, Physical Review A 106, 062614 (2022), doi:10.1103/PhysRevA.106.062614.
• [72] S. Morita and H. Nishimori, Mathematical foundation of quantum annealing, Journal of
Mathematical Physics 49(12) (2008), doi:10.1063/1.2995837.
• [73] S. Morita and H. Nishimori, Convergence theorems for quantum annealing, Journal of
Physics A: Mathematical and General 39(45), 13903 (2006).
• [74] A. del Campo, Shortcuts to adiabaticity by counterdiabatic driving, Physical Review
Letters 111, 100502 (2013), doi:10.1103/PhysRevLett.111.100502.
• [75] I. ˇCepait˙e, A. Polkovnikov, A. J. Daley and C. W. Duncan, Counterdiabatic optimized
local driving, PRX Quantum 4, 010312 (2023), doi:10.1103/PRXQuantum.4.010312.
• [98] T. Hatomura and K. Takahashi, Controlling and exploring quantum systems by algebraic
expression of adiabatic gauge potential, Physical Review A 103, 012220 (2021),
doi:10.1103/PhysRevA.103.012220.
• [108] E. D. C. Lawrence, S. F. J. Schmid, I. ˇCepait˙e, P. Kirton and C. W. Duncan, A
numerical approach for calculating exact non-adiabatic terms in quantum dynamics, SciPost
Physics 18, 014 (2025), doi:10.21468/SciPostPhys.18.1.014.
We hope that these changes adequately address Referee 1’s concerns and that our paper can thus
be accepted for publication.
Best regards,
the Authors

---

## Round 2 · Referee Report · Anonymous (Referee 2) · 2025-3-27

Strengths
- the paper has a good presentation of both no-core shell model and quantum annealing protocol suitable for a general audience
- the authors study a reasonably large variety of nuclear systems
Weaknesses
- the results presented in the paper are obtained from exact classical simulation, it is therefore difficult to asses the practical viability of the proposed method on real devices affected by noise
- the estimated gate complexity of the proposed algorithm seems to suggest that the protocol might require resources beyond those expected in NISQ devices. If the algorithm were to be run on fault-tolerant quantum computers the number of required CNOT will be of little consequence and a study of the non-Clifford resource needed should be carried out instead
Report
The present manuscript describes an application of Quantum Annealing to the problem of preparing ground states of nuclear systems described using No-Core Shell Model. The authors present results obtained with classical simulations on a variety of nuclei and estimate the gate complexity of the algorithm on digital quantum platforms. The paper is well written and the results interesting, however it does not appear to satisfy any one of the expectations required for publication in SciPost Physics: the QA annealing approach is well known and the novel elements proposed here, such as the choice of driving Hamiltonian, do not seem to be general or ground-breaking enough. I believe that the work is solid but that it might be more suitable for a different Journal.
Below my main comments on the manuscript
-
The numerical results in Sec.5 are certainly interesting but in part simply confirm previous expectations, like the 1/\Delta^2 scaling of the evolution time, while it appears difficult to extrapolate the other observations on the observed performance of different systems to other systems that have not been studied. For instance, would it be possible to leverage the current results to, at least approximately, predict the cost of annealing on larger Hilbert spaces for these nuclei? What about with other nuclear interactions?
-
Right after Eq.(18) the authors suggest that choosing an initial product state that minimizes the energy of the target Hamiltonian H_T may be advantageous. Of the two provided references the first has some compilation error (shows up as a ?) while Ref.[84] is not available yet. Could the authors briefly explain the reasons for the possible advantage of using low energy initial states?
-
In the conclusions at the end of Sec.6 the authors suggest that QA would scale better than ADAPT-VQE as a function of D under the assumption that the number of steps required in QA is independent on D. I think this statement needs some further clarification. As expected from adiabatic algorithms, and confirmed by the results shown in Fig.5, the number of steps of evolution will scale as the inverse gap squared. The expectation is then that: if the smallest gap \Delta depends only polynomially on D, then the total cost of QA (cost per step times number of steps) will also scale polynomially.
-
There seems to be a small mistake right before Eq.(22): the authors say that D CNOT are needed to implement the exponential of a Pauli string of length D/2. I tried to look for this result in Ref.[39] but couldn't find it. In general however, it is well known that an exponential of a length N Pauli string requires a staircase of 2(N-1) CNOT instead (see e.g. Fig.8 of Ref.[39]). In the case discussed in the manuscript the CNOT cost should then be D-2 instead of D.
-
The explicit definitions of \mathcal{B} and \mathcal{B}_0 in Figure 1 should probably be in the main text next to where these space are defined instead of in a Figure.
-
In the conclusions the authors say that the QA approach scales polynomially in the number of many-body states. I believe they meant to say single-particle states instead.
Recommendation
Accept in alternative Journal (see Report)

---

## Round 3 · Referee Report · Anonymous (Referee 1) · 2025-5-19

Strengths

This article 1) clearly discuss a new strategy to address nuclear structure using quantum annealing 2) it also gives discussions and issues that might be important for future applications.

Weaknesses

1) One weakness is that some recent progress in quantum annealing are not really used. 2) The article is still rather theoretical without touching the difficulties associated to real devices.

Report

In this new version, the authors have responded to my main concern regarding the proper citation of recent works where adiabatic methods and related aspects are discussed. I reiterate my comments regarding this work. This article is one of the first where annealing is considered for solving the shell model. As such, it represents a valuable discussion of various aspects that are of importance for current and future applications. I again recommend it for publication.

Requested changes

No change requested

Recommendation

Publish (meets expectations and criteria for this Journal)

---

## Round 3 · Referee Report · Anonymous (Referee 2) · 2025-6-3

Report

The authors have addressed all of my previous comments in the revised version. They also made a stronger case about the novelty of the present work, in particular regarding the suitability of adiabatic state preparation for nuclear ground states in the shell model thanks to the favorable gaps. I believe that the extensive numerical evidence they provide is a useful stepping stone towards practical implementation of the nuclear shell model on quantum devices.

Recommendation

Publish (meets expectations and criteria for this Journal)

---

## Round 3 · Author Response

Dear Editor, We thank Referees 1 and 2 for their careful reading of our manuscript and for their constructive comments and suggestions, which we have incorporated into the improved version we are resubmitting. We are especially thankful to Referee #1, for recommending publication of the manuscript in SciPost Physics with minor revision. However, we respectfully disagree with the editorial decision to resubmit our work to the journal SciPost Core because “[it] lacks the novelty e.g. of a new aspect to the method or actual implementation on hardware, does not meet the criteria for the flagship journal.” While we agree that our work solves a nuclear shell-model Hamiltonian by a classical simulation of quantum annealing, in our view our manuscript opens the way for nuclear structure studies using quantum annealers, and such pioneer proof of principle study is a necessary starting point for future implementations on quantum devices. Indeed, the novelty of our work is supported by the report of Referee 1, which highlights that “As far as I know, this is one of the first attempts made to assess the quality of such a strategy for the nuclear shell model. From that point of view, the present work contains many interesting information and quantitative studies. It might even be seen as a milestone before trying to attack applications on real quantum platforms”. Along the same lines, we would like to emphasize that, as indicated in our submission, in our opinion our manuscript meets two of the acceptance criteria (at least one required) for publication in SciPost Physics: 1. Provide a novel and synergetic link between different research areas in particular with the first application of quantum adiabatic computing to address a nuclear structure problem, bridging two fields that, to our knowledge, were previously disconnected; and 2. Open a new pathway in an existing or a new research direction, with clear potential for multi-pronged follow-up work as our manuscript demonstrates that our framework is promising to obtain at least the ground states of light and medium-mass nuclei, thereby setting the basis for future investigations on the practical implementation of general fermionic Hamiltonians using quantum adiabatic computation. In particular, our work opens multiple research avenues for investigation, such as: alternative fermion-to-qubit mappings to encode non-local nuclear Hamiltonians in current quantum annealers; synergetic developments in the quantum annealing community to explore the potential of next-generation annealers to meet the requirements to simulate nuclear structure Hamiltonians; or more efficient quantum annealing protocols—driver Hamiltonians, annealing schedule—to reduce quantum resources. Additionally, our work provides a new framework with potential to study nuclear dynamics, including nuclear processes mediated by the weak and electromagnetic interactions, as well as beyond-standard-model physics involving atomic nuclei. For these reasons, we kindly invite you to reconsider our resubmission for publication in SciPost Physics. Along with this letter, we provide detailed responses to each of the points raised by Referees 1 and 2. All modifications have been incorporated into the revised manuscript, which has been improved accordingly. Sincerely, E. Costa, A. Perez-Obiol, J. Menendez, A. Rios, A. Garcia-Saez and B. Julia Diaz

---

## Round 3 · List of Changes

Referee 2, Comment 1: “The paper is well written and the results interesting, however it does not appear to satisfy any one of the expectations required for publication in SciPost Physics: the QA annealing approach is well known and the novel elements proposed here, such as the choice of driving Hamiltonian, do not seem to be general or ground-breaking enough.”

Authors Response: We kindly disagree with Referee 2 on this point. In our opinion, our manuscript meets two of the acceptance criteria (at least one required) for publication in SciPost Physics:

"Provide a novel and synergetic link between different research areas" Our work is the first application of quantum adiabatic computing to address a nuclear structure problem. It provides a clear original synergy between the so-far disjoint research areas of nuclear physics and adiabatic quantum computing. Our work bridges the gap between these two fields and has the potential for several new synergetic efforts.
2." Open a new pathway in an existing or a new research direction, with clear potential for multipronged follow-up work" Our manuscript demonstrates that the quantum adiabatic computing framework is a promising avenue to obtain at least the ground states of light and medium-mass nuclei, thereby setting the basis for future investigations on the practical implementation of general fermionic Hamiltonians using quantum adiabatic computation. While some of the previous applications of many-body systems in an adiabatic setting exploit intrinsic symmetries (eg locality), the nuclear shell model is a complex, realistic non-local hamiltonian. Our work, however, opens a new way to exploiting adiabatic quantum computing in addressing these physical models. In particular, our work opens multiple research avenues for investigation, such as: alternative fermion-to-qubit mappings to encode non-local nuclear Hamiltonians in current quantum annealers; synergetic developments in the quantum annealing community to explore the potential of next-generation annealers to meet the requirements to simulate nuclear structure Hamiltonians; or more efficient quantum annealing protocols—driver Hamiltonians, annealing schedule—to reduce quantum resources. Additionally, our work provides a new framework with potential to study nuclear dynamics, including nuclear processes mediated by the weak and electromagnetic interactions, as well as beyond-standard-model physics involving atomic nuclei.

In order to emphasize these merits, we have added the following sentences to the paper. For Criterion 1 (Provide a novel and synergetic link between different research areas), we add in the first paragraph of the Conclusions: • “Our work is, to our knowledge, the first application of quantum adiabatic computing to address a nuclear structure problem, enabling a clear synergy between two so-far disconnected research areas.” About Criterion 2 (Open a new pathway in an existing or a new research direction, with clear potential for multi-pronged follow-up work), we add in the last paragraph of the Conclusions: • “The quantum adiabatic framework for nuclear structure presented here opens several new potential research avenues at the interface between these two research areas.” .

Referee 2, Comment 2: “The numerical results in Sec.5 are certainly interesting but in part simply confirm previous expectations, like the 1/Δ^2 scaling of the evolution time, while it appears difficult to extrapolate the other observations on the observed performance of different systems to other systems that have not been studied. For instance, would it be possible to leverage the current results to, at least approximately, predict the cost of annealing on larger Hilbert spaces for these nuclei? What about with other nuclear interactions?”

Authors Response: We appreciate the point raised by Ref. 2 regarding the generalization of our results to larger Hilbert spaces and different nuclear interactions. In our study we have identified two important features: i) using a linear schedule, we find τ ∝ Δ^−2, in agreement with the approximate adiabatic law; and ii) the instantaneous energy gap throughout most of the evolution is larger than the gap of the target Hamiltonian. We expect these features to be general, and independent of the size of the Hilbert space used. Indeed, there is no clear correlation between the dimension of the Hilbert space and the energy error for the nuclei studied so far, as illustrated in Fig. [see attached file]. Moreover, we have already observed that nuclei with a number of particles that can fill a low energy subshell (28^Si,32^Ar,12^C), have a large gap between the ground state and the first excited state along the adiabatic evolution. Since this property is established by the structure of these nuclei, we do not expect significant changes by calculating in a larger Hilbert space. In general, larger Hilbert spaces will mostly affect the physics of higher-energy nuclear states, which should not affect much the energy gap. Likewise, we do not expect any qualitative change by the use of other nuclear Hamiltonian. In our work we already explore two different nuclear interactions for p- and sd-shell nuclei. These are phenomenological in nature, but the systematics of our results across the different mass numbers suggest that our results do not depend significantly on nuclear interactions. Also, since the experimental low-energy structure of the studied nuclei is well captured by the standard nuclear Hamiltonians we use, we do not expect any notable changes by using a different nuclear Hamiltonian, which should also reproduce well the same low-energy nuclear structure data. In order to clarify this aspect, we have added a paragraph at the end of Section 5: • “Finally, we expect these main features of the QA protocol to be robust with respect to an enlargement of the Hilbert space for the same nucleus, which should mostly affect higher-energy excited states leaving the energy gap relatively stable. Likewise, we also anticipate similar results from using different nuclear Hamiltonians, as long as they capture well the experimental low-energy structure of the nuclei studied in this work. Indeed, we already explore two different sets of nuclear interactions for p- and sd-shell nuclei. These are phenomenological in nature, but the systematics of our results across the different mass numbers suggest that our results do not depend significantly on the nuclear interaction used.”

Referee 2, Comment 3: “Right after Eq.(18) the authors suggest that choosing an initial product state that minimizes the energy of the target Hamiltonian H_T may be advantageous. Of the two provided references the first has some compilation error (shows up as a ?) while Ref.[84] is not available yet. Could the authors briefly explain the reasons for the possible advantage of using low energy initial states?”

Authors Response: First, we apologize for the the typo in the citation, which we have corrected. The corresponding reference is [90] in the improved manuscript. Second, addressing the question, our rationale is that the lowest-energy basis state can be advantageous because one may naively expect it to have a good overlap with the ground state of the target Hamiltonian. Moreover, starting from a low-energy initial state leads to a smaller variation in the ground-state energies along the QA protocol, which may be useful for experimental realizations. We have clarified this aspect by modifying the sentence to • “A lower energy closer to the ground-state energy of the target Hamiltonian may be advantageous for a potential QA realization, although we emphasize that it does not guarantee faster or more efficient convergence than other initial states [90-91]”.

Referee 2, Comment 4: “In the conclusions at the end of Sec.6 the authors suggest that QA would scale better than ADAPTVQE as a function of D under the assumption that the number of steps required in QA is independent on D. I think this statement needs some further clarification. As expected from adiabatic algorithms, and confirmed by the results shown in Fig.5, the number of steps of evolution will scale as the inverse gap squared. The expectation is then that: if the smallest gap Δ depends only polynomially on D, then the total cost of QA (cost per step times number of steps) will also scale polynomially.”

Authors Response: We fully agree on this comment by Referee 2. It is well known that the energy gap between the ground state and the first excited state of the nuclear shell model Hamiltonians scales polynomially with the number of degrees of freedom. Since the minimum gap follows the energy gap of these Hamiltonians, we expect that also Δ scales polynomially. We have added the following sentence in the last paragraph of Sec. 6, • “In addition to that, we expect that the number of time steps also scales polynomially with D, since τ ∝ Δ^−2, and Δ approaches the energy gap between the ground state and the first resonance state in H_T , that for nuclei scales polynomially with D [13,14]. In this perspective, our approach may scale better than ADAPT-VQE, which requires an exponentially growing number of CNOTs proportional to the number of many-body states [44, 52].” and we have also included a new sentence in the thrid paragraph of the Conclusions, • “it scales polynomially with respect to the number of nucleon modes because the scaling of the energy gap of the target Hamiltonian is also polynomial.”

Referee 2, Comment 5: “There seems to be a small mistake right before Eq.(22): the authors say that D CNOT are needed to implement the exponential of a Pauli string of length D/2. I tried to look for this result in Ref.[39] but couldn’t find it. In general however, it is well known that an exponential of a length N Pauli string requires a staircase of 2(N-1) CNOT instead (see e.g. Fig.8 of Ref.[39]). In the case discussed in the manuscript the CNOT cost should then be D-2 instead of D.”

Authors Response: We appreciate Referee 2 for catching the error. We have corrected the mistake and now we estimate correctly the number of CNOT gates required, which reads... • “the number of CNOT gates required to exponentiate a Pauli operator of length D/2 is (D − 2) [102]. Therefore, the number of CNOTs for a unitary gate is NCNOT ∝ D^4(D − 2)′′

Referee 2, Comment 6: “The explicit definitions of B and B0 in Figure 1 should probably be in the main text next to where these space are defined instead of in a Figure.”

Authors Response: We have followed the Referee’s recommendation and have removed the definitions on Figure 1 and added them in the main text above Eq.(18), which now reads: • “Finally, it is noteworthy that our choice for the initial state corresponds to the minimum energy Slater determinant in B0 = {|s⟩ ∈ B | M |s⟩ = 0}, where B indicates the many-body basis [...] ”

Referee 2, Comment 7: “In the conclusions the authors say that the QA approach scales polynomially in the number of many-body states. I believe they meant to say single-particle states instead.”

Authors Response: That is correct, we thank the Referee for spotting this error. We corrected the mistake changing the sentence in the following way: • “ [...] it scales polynomially with respect to the number of nucleon modes [...]” and we have also corrected a similar inaccuracy in the abstract: • “[...] is polynomial in the single-particle basis size [...]”.

Referee 1, Comment and Requested Changes 1:
“With this last argument, I estimate that the present article should be published provided that some
more discussion is made on the recent efforts made to perform calculations using real quantum
platforms, at least to better present the work as an academic intermediate step and quote the ongoing
works that are made to fill the gap between this type of ideal work the progress that are made to
apply quantum annealing today on real quantum processors. [...] Better discuss the current work
on quantum annealing regarding the criteria of convergence and the different methods to accelerate
the adiabatic approach.”

Authors Response:
We thank Referee 1 for the suggestion. We certainly agree it is important to connect our work with
current efforts to perform simulations in real quantum annealers, including systems where classical
simulations are not feasible. To this end, we have added a new sentence in the Introduction:
• "Several studies have been also focused on both the criteria of convergence [71–73] and the
speed-up of the QA protocol via counteradiabatic drivings [74, 75]."
and an almost entire new paragraph in the Conclusion, indicating the various potential research
directions:
• "Future investigations could explore criteria of convergence of the QA protocol without recurring
to classical simulations, based on the form of the scheduler or the structure of the driver
Hamiltonian [71–73]. Moreover, it is possible to explore ways to speed up the QA protocol by
using counteradiabatic drivings [74, 75], studying the structure of the adiabatic gauge potential
[98, 108] or using quantum optimal control techniques on the scheduler [70]."
which include the following additional references incorporated into the manuscript:
• [71] Y. Kimura and H. Nishimori, Convergence condition of simulated quantum annealing for
closed and open systems, Physical Review A 106, 062614 (2022), doi:10.1103/PhysRevA.106.062614.
• [72] S. Morita and H. Nishimori, Mathematical foundation of quantum annealing, Journal of
Mathematical Physics 49(12) (2008), doi:10.1063/1.2995837.
• [73] S. Morita and H. Nishimori, Convergence theorems for quantum annealing, Journal of
Physics A: Mathematical and General 39(45), 13903 (2006).
• [74] A. del Campo, Shortcuts to adiabaticity by counterdiabatic driving, Physical Review
Letters 111, 100502 (2013), doi:10.1103/PhysRevLett.111.100502.
• [75] I. ˇCepait˙e, A. Polkovnikov, A. J. Daley and C. W. Duncan, Counterdiabatic optimized
local driving, PRX Quantum 4, 010312 (2023), doi:10.1103/PRXQuantum.4.010312.
• [98] T. Hatomura and K. Takahashi, Controlling and exploring quantum systems by algebraic
expression of adiabatic gauge potential, Physical Review A 103, 012220 (2021),
doi:10.1103/PhysRevA.103.012220.
• [108] E. D. C. Lawrence, S. F. J. Schmid, I. ˇCepait˙e, P. Kirton and C. W. Duncan, A
numerical approach for calculating exact non-adiabatic terms in quantum dynamics, SciPost
Physics 18, 014 (2025), doi:10.21468/SciPostPhys.18.1.014.

---

## Editorial Decision

published